

# Food addiction and the physical and mental health status of adults with overweight and obesity

Magdalena Zielińska[1], Edyta Łuszczki[1], Anna Szymańska[2] and Katarzyna Dereń[1]

[1] Institute of Health Sciences, College of Medical Sciences, University of Rzeszów, Rzeszów, Poland
[2] Rzeszów Association for Disabled and Autistic Children SOLIS RADIUS, Rzeszów, Poland

## ABSTRACT

**Background:** Overweight and obesity now affect more than a third of the world's population. They are strongly associated with somatic diseases, in particular increasing the risk of many metabolic and cardiovascular diseases, but also with mental disorders. In particular, there is a strong association between obesity and depression. As a result, more attention is paid to the neurobiological, behavioural, and psychological mechanisms involved in eating. One of these is food addiction (FA). Research comparing lifestyle elements, physical and mental health problems of excess body weight and individuals with FA is limited and has focused on younger people, mainly students. There is also a lack of studies that relate actual metabolic parameters to FA. To better understand the problem of FA also in older adults, it is important to understand the specific relationships between these variables.
**Methods:** A cross-sectional survey was conducted with 172 adults with overweight and obesity (82% female) aged 23–85 years. The mean age of all subjects was $M = 59.97$ years ($SD = 11.93$), the mean BMI was $M = 32.05$ kg/m$^2$ ($SD = 4.84$), and the mean body fat was $M = 39.12\%$ ($SD = 6.48$). The following questionnaires were used: Food Frequency Questionnaire-6 (FFQ-6), Global Physical Activity Questionnaire (GPAQ), Three Factor Eating Questionnaire-R18 (TFEQ-R18), Yale Food Addiction Scale 2. 0 (YFAS 2.0), Zung Self-Rating Depression Scale (SDS). Body composition, anthropometry, fasting glucose, lipid profile, and blood pressure were measured.
**Results:** A total of 22.7% of participants with overweight and obesity had symptoms of depression according to the SDS, and 18.6% met the criteria for FA according to YFAS 2.0. FA was statistically significantly more common among people up to 50 years. BMI, body fat mass, diastolic blood pressure and sedentary behaviour were statistically significantly higher in people with FA symptoms. Those who were sedentary for 301–450 min per day were significantly more likely to have depressive symptoms, and those who were sedentary for more than 450 min per day were significantly more likely to have FA symptoms.
**Conclusions:** Our findings complement the current literature on FA, particularly in older adults and metabolic parameters, and suggest further research directions. Although our cross-sectional study design does not allow causal interpretations, increasing physical activity appears to be particularly important in the management of people with overweight or obesity and FA. This may be even more important than

Corresponding author
Magdalena Zielińska,
mazielinska@ur.edu.pl

for people with depression alone, but future research is needed to explore these relationships further.

## INTRODUCTION

Overweight and obesity now affect more than a third of the world's population (*Hruby & Hu, 2015*). According to the latest data, adult obesity rates more than doubled between 1990 and 2022 (*NCD Risk Factor Collaboration (NCD-RisC), 2024*). They are strongly associated with somatic diseases, in particular increasing the risk of many metabolic and cardiovascular diseases, but also with mental disorders (*Chu et al., 2019*; *Yang, Liu & Zhang, 2022*). This has huge implications for the quality and duration of life of individuals and for healthcare systems (*Lin & Li, 2021*). An obesogenic environment, the increasing availability of highly processed foods, adverse lifestyle changes, and reduced physical activity (PA) exacerbate the problem (*Wadden, Tronieri & Butryn, 2020*). In addition to an unhealthy lifestyle, many different biological, psychological, behavioural and social factors contribute to overweight and obesity (*Firth et al., 2020*).

Consequently, more attention is paid to the neurobiological, behavioural, and psychological mechanisms involved in eating (*Becetti et al., 2023*). One of these is food addiction (FA), which is currently the subject of intense discussion and research despite the lack of a standard definition and consensus in the scientific community (*Hauck, Cook & Ellrott, 2020*; *Vasiliu, 2022*). The Yale Food Addiction Scale 2.0 (YFAS 2.0) is most commonly used to assess the presence of symptoms of FA (*Gearhardt, Corbin & Brownell, 2016*). FA is a condition characterised by a lack of control over food intake, strong appetite, excessive food consumption despite negative health or social consequences, and repeated unsuccessful attempts to control intake (*Gearhardt & Hebebrand, 2021*). This construct was developed to help understand the dysfunctional eating patterns observed in patients with overweight, obesity, and eating disorders (*Imperatori et al., 2015*; *Jiménez-Murcia et al., 2019*).

In recent years, FA has been observed in patients with overweight and obesity, with a prevalence of approximately 25% (*Pursey et al., 2014*; *Imperatori et al., 2017*). Furthermore, FA has been positively associated with body fat level and the Body Mass Index (BMI) (*Gearhardt, Boswell & White, 2014*; *Pedram et al., 2013*). FA is likely to be an important factor in the development of obesity in humans and is associated with the severity of obesity (*Lerma-Cabrera, Carvajal & Lopez-Legarrea, 2016*). The prevalence of co-occurrence of obesity and FA is estimated to be between 15% and 57% (*Minhas et al., 2021*). A high prevalence of FA is found more frequently in clinical groups, that is, eating disorders, mainly bulimia, binge eating disorder (BED) (70–90%), candidates for bariatric surgery (14–58%) (*Schankweiler et al., 2023*; *Salehian et al., 2023*). The diagnostic criteria for FA are based on the criteria for substance use disorders in the Diagnostic and Statistical Manual of Mental Disorders, Fifth Edition (DSM-V), but are not included in this

classification or in the International Classification of Diseases (ICD-11) (*American Psychiatric Association, 2013*). There may be a subgroup of people with overweight and obesity who meet the criteria for FA and exhibit problematic eating behaviours, but who may not receive appropriate treatment (*Meule, 2019*). If so, it is important to understand this subgroup to develop targeted interventions.

It has been suggested that addictive consumption of highly processed, highly palatable foods with increased fats and refined carbohydrates may influence weight gain and obesity (*De Almeida, Kamath & Cabandugama, 2022*). This type of food releases dopamine (DA) in an area of the brain called the nucleus accumbens (NAcc) and alters its transmission (*de Macedo, de Freitas & da Silva Torres, 2016*). Furthermore, both obesity and addiction are associated with a reduction in the number of dopamine D2 receptors in the brain, suggesting that people with obesity may be less sensitive to food-related reward stimuli (*Wang et al., 2001*). Dopaminergic pathways are likely to be involved in both addictive processes and depressive symptoms, and this may be the basis for their co-occurrence (*Grajek et al., 2022*).

The process of eating not only provides nutrients and energy, but also influences mental health through the regulation of eating in relation to the experience of pleasure and the activation of the reward centre (*Campos, Port & Acosta, 2022*). Both homeostatic and hedonic processes are involved in co-occurring FA, excessive body weight, and depressive symptoms (*Burrows et al., 2018*). Studies suggest that there are complex interactions between FA and levels of depression symptoms. A meta-analysis of longitudinal studies by Luppino et al. showed that depression is a predictor of the development of obesity (*Luppino et al., 2010*). One of the behavioural mechanisms linking depression and the subsequent development of obesity has been suggested to be eating in response to negative emotions (*Konttinen et al., 2019*). Emotional eating tends to cooccur with uncontrolled eating, and these two forms of eating are distinguished by different factors and are related to other emotional aspects, such as depressive feelings (*Benbaibeche et al., 2023*). FA has been associated with emotional eating (*Schankweiler et al., 2023*; *Escrivá-Martínez et al., 2023*). Furthermore, the prevalence of eating disorders and psychiatric symptoms, as well as depression, is also higher in participants undergoing weight loss interventions (*Davis, 2015*; *Weinberger et al., 2018*). According to *Burmeister et al. (2013)* the presence of FA may be an important factor influencing the effectiveness of weight loss programmes and the overall mental and social health of people trying to lose weight.

People classified as with FA were shown to consume a higher part of the energy requirements of ultraprocessed food (UPF) (*Whatnall et al., 2022*). Studies have also shown that addictive UPF consumption can explain, at least in part, the observation that people with obesity and FA report a range of comorbidities compared to people with obesity without FA, including poorer cardiovascular health indicators and increased prevalence and severity of psychiatric disorders (such as depression) (*Pagliai et al., 2021*). There has been shown to be a link between depression and metabolic syndrome on various levels, especially obesity (*Al-Khatib et al., 2022*). Importantly, metabolic disorders have been linked to an increased risk of coronary heart disease, myocardial infarction, heart failure, hypertension and stroke (*Wang et al., 2023*). In addition, people with metabolic

disorders have a higher risk of mortality (*Li et al., 2023*; *Käräjämäki et al., 2022*). FA is also strongly associated with the presence of type 2 diabetes, according to current data published in the literature (*Lavielle et al., 2023*; *Horsager et al., 2023*). Type 2 diabetes, on the other hand, is also closely related to mental health (*Mukherjee & Chaturvedi, 2019*).

Interestingly, it has been suggested that the frequency and duration of PA, sedentary behaviour (SB), and sleep time are associated with FA (*Li et al., 2018*). People with FA had significantly less PA and reported significantly more symptoms of poorer sleep quality. Existing studies showed that the presence of FA is not influenced by the level of PAy; however, participants who reported a high level of PA showed more symptoms of food addictive behaviour than those with low and moderate PA (*Bailey et al., 2017*). A study among adults with obesity found that higher levels of PA were associated with low levels of FA (*Brytek-Matera et al., 2021*).

Research comparing lifestyle elements, physical and mental health problems of excess body weight and FA individuals is limited and has focused on younger people, mainly students (*Kim, Kang & Shin, 2023*; *Rivera-Mateos & Ramos-Lopez, 2023*; *Bartschi & Greenwood, 2023*). There is also a lack of studies that relate actual metabolic parameters to FA. To better understand the problem of FA, not only in young adults but also in older adults, it is important to understand the specific relationships between these variables. In this regard, it is of essential interest to further understand the clinical significance of FA and to identify possible therapeutic targets.

The study had two main objectives. The first was to determine the differences between FA and NFA concerning age, eating habits and physical health (comorbidities, metabolic and anthropometric parameters) of adults with overweight and obesity attending a private centre offering weight loss holidays for adults. Given the positive association between FA and depression, the second objective was to examine the associations between FA and depressive symptoms and factors such as BMI, eating behaviours and PA and SB.

Therefore, the study distinguished between two areas of research and formulated the following hypotheses:

## Analysis of factors associated with FA

H1. There are differences in the prevalence of FA according to age.

H2. There is a significant difference in the frequency of consumption of different products in subgroups between the subgroup of FA in contrast to non-food-addicted (NFA) participants.

H3. There is a significant relationship between FA and the number of comorbidities and metabolic parameters such as blood pressure, total cholesterol (TC), high-density lipoprotein (HDL), low-density lipoprotein (LDL), and triglycerides (TG), fasting glucose level and anthropometric results.

## Relationship between FA and depressive symptoms

H4. There is a significant relationship between BMI, eating behaviours, PA and SB, and symptoms of FA and depression.

## MATERIALS AND METHODS

### Participants

The cross-sectional study was conducted in a private resort in Poland that organises weight loss holidays for adults. The survey was conducted between July 2023 and October 2023, during the period when the resort organized weight loss holidays. A total of 204 people who approached the weight reduction centre agreed to participate in the study. The study excluded subjects aged <18 and >85 years, with a BMI < 25.0 kg/m$^2$, current diagnoses of psychiatric disorders, ie schizophrenia, epilepsy, dementia. Twenty-seven participants did not meet the inclusion criteria. Furthermore, incomplete results were discarded ($n = 3$) and two participants withdrew from the study. The final sample consisted of $N = 172$ individuals with overweight or obesity.

### Procedures

All measurements and questionnaires were taken before any changes were made in diet or lifestyle. The first day in the centre, the day of the study, was a day off from the scheduled activities, so that the participants could take their time to fully participate in the study. A multidisciplinary research team, including a nurse, psychologist, dietitian, and psychodietitian, explained the details and procedures of the study and conducted it in the morning on an empty stomach. A doctor was also present at the centre on the day of the study. All stages of the process were carefully optimised to ensure that participants had adequate breaks between each stage of the survey to ensure the highest possible data quality and minimise possible fatigue of respondents. This study follows the guidelines for cross-sectional surveys (*von Elm et al., 2008*).

### Measures

#### Sociodemographic variables and covariates

A questionnaire was used to collect general information. Included questions on gender, date of birth, level of education, place of residence, medical history, history of eating disorders, current use of pharmacotherapy, occupational activity, current smoking, amount of sleep (*Jezewska-Zychowicz et al., 2018*).

#### Frequency of food consumption

The habitual food intake of each participant was assessed using a qualitative Food Frequency Questionnaire (FFQ-6®), which includes 62 different food items representing eight main food groups (*Wądołowska, 2005*). The respondents were given the opportunity to select one of six categories the frequency of food consumption in the past 12 months, ranging from one *(never or almost never)* to six *(several times a day)*. As the authors of the questionnaire point out, the data obtained can be used to identify people with different levels of consumption of certain products and/or to identify characteristic patterns of food consumption. When the questionnaire was validated, the average correlation coefficients for the frequency of consumption of individual products were r = 0.78 (95%CI [0.73–0.83]) and for the quantity of products were r = 0.76 (95%CI [0.71–0.81]) (*Wądołowska, 2005*).

FFQs are often used for dietary evaluation because they are practical, simple, quick to complete, relatively inexpensive, require less effort from the respondent, and are more representative of the average diet than other methods (*Cade et al., 2002*).

### Eating behaviours

For the assessment of problematic eating behaviour, the revised version of the Three Factor Eating Questionnaire 18-items (TFEQ-R18) is used in the Polish adaptation of the Eating Disorders Questionnaire (*Karlsson et al., 2000*; *Brytek-Matera, Rogoza & Czepczor-Bernat, 2017*). TFEQ-R18 assesses cognitive restraint (deliberate adherence to inappropriate and/ or restrictive diets), uncontrolled eating (behaviours that indicate a lack of control over eating, even when not physically hungry or eating much faster than usual) and emotional eating (is based on the psychosomatic theory that some people are unable to distinguish between hunger and other bodily arousal *e.g.*, emotions) (*Anglé et al., 2009*). Thirteen items are rated on a four-point scale from one *(does not describe me at all)* to four *(describes me accurately)*, four have other responses on a four-point scale, and one question is rated on an 8-point scale (from one–no abstinence to eight–total abstinence). The raw scale scores are converted to a 0–100 scale (((raw score – lowest possible raw score)/(possible range of raw scores)) * 100) (*Paans et al., 2018*). The reliability of the TFEQ scale using the Cronbach alpha method was 0.607 in our group. Questions 1, 9, 10, 14 and 17 were negatively correlated with the total score. After recoding these questions, an alpha of 0.868 was obtained. The authors have permission to use this instrument from the copyright holders.

### Physical activity and sedentary behaviour

The Global Physical Activity Questionnaire (GPAQ), developed by the World Health Organization (WHO) in 2002 was used to measure PA and SB (*Bull, Maslin & Armstrong, 2009*; *Bergier, Wasilewska & Szepeluk, 2019*). The GPAQ is a synthetic combination of the short and long versions of the IPAQ. The GPAQ consists of 16 questions divided into four parts. These relate to time spent in PA at work, time spent in PA during a typical week (walking and cycling for at least 10 min without a break), time spent in leisure time, and the fourth section relates to SB. The GPAQ data were processed and analysed using the WHO GPAQ analysis guide (*World Health Organization, 2012*). The total PA per week was calculated using metabolic equivalents (MET) of activity levels according to WHO recommendations (*Rodríguez-Romo et al., 2022*). Participants with low levels of PA were defined as not sufficiently active. Those considered sufficiently active were those classified as having moderate and high levels of PA. People who are insufficiently active are those who do not meet the WHO recommendations for adults of at least 150 min of moderate-intensity PA, or 75 min of vigorous-intensity PA, or an equivalent combination of moderate and vigorous-intensity PA, for a total of at least 600 min of MET minutes (*World Health Organization, 2020*). Reliability tests of the GPAQ in nine countries showed moderate to significant reliability (Kappa 0.67 to 0.73; Spearman's rho 0.67 to 0.81) (*Bull, Maslin & Armstrong, 2009*). The authors have permission to use this instrument from the copyright holders.

### Food addiction

The presence of symptoms of FA was measured using YFAS 2.0. Gearhardt, Corbin and Brownell scale in Polish adaptation (*Poprawa et al., 2020*). YFAS 2.0 is a self-report questionnaire that has been validated to assess addictive eating behaviours in the last 12 months. Each of the 35 questions contains answers on a scale from zero *(never)* to seven *(every day)*. The results obtained can be differentiated in terms of the severity of FA into: one–mild, two–moderate, three–severe. Based on the 11 diagnostic criteria for SUD in DSM-5 (*e.g.*, craving, tolerance, or withdrawal), a symptom score of 0 to 11 is calculated, reflecting the severity of FA symptoms. In addition, clinically significant impairment or distress due to eating behaviour is evaluated. The presence of no more than one symptom or the absence of symptom 12 *(substance use causes serious problems or distress)* was classified as non-dependence. FA is diagnosed when two or more symptoms are present together with symptom 12 (as mild dependence), 4–5 symptoms together with symptom 12 were classified as moderate dependence, and the presence of more symptoms together with 12 was classified as severe dependence (*Gearhardt, Corbin & Brownell, 2016*). The Cronbach's α reliability analysis for YFAS 2.0 in this sample of respondents was 0.953, indicating a very high internal consistency of the scale used. The authors have permission to use this instrument from the copyright holders.

### Self-rated depression

The self-rated depressive symptoms were measured using the Zung Depression Self-Rating Scale (SDS) (*Zung, 1965*). The SDS is the commonly used self-rating scale (next to the Beck Depression Scale–BDI). The high reliability of the scale confirmed in research justifies its use (*Dunstan, Scott & Todd, 2017*; *Velescu et al., 2024*). It includes 20 symptoms divided into four groups: mood disturbances, physiological functions, psychomotor activity, and psychiatric symptoms. The scale uses affirmative sentences (10 describing pathological phenomena, 10 referring to the presence of normal features). Some items are reverse scored (*i.e.*, they go from four down to one). The scale produces raw scores between 20 and 80, however these values can be converted to Index Scores (from 25 to 100) by the process of multiplying by 1.25 (*Zung, 1965*).

A cut-off point ≥50 for raw scores is recommended (*Dunstan & Scott, 2018*; *Velescu et al., 2024*). For the 50-point cut-off applied by many researchers, sensitivity is 78.9% and specificity is 83.7% (*Dunstan & Scott, 2019*). Based on the findings, the use of an SDS raw score of 50 as the cut-off point. An SDS score of 50 or higher may suggest the presence of clinically significant symptoms of depression that may require further evaluation and therapeutic intervention. In this study, the number of points scored indicates <50 points—no depressive symptoms, 50–59 points—those with mild to moderate depression, 60–69 points—those with moderate to severe depression, ≥70 points—those with severe to very severe depression. The SDS scale was also validated to identify clinically significant symptoms of depression in the older population compared to the BDI-21 scale (*Jokelainen et al., 2019*). The reliability analysis for the SDS in the study group was Cronbach's α = 0.870, indicating the internal consistency of the scale used.

### Waist and hip circumference

Waist and hip circumferences were measured twice with a 205 cm SECA tape measure to an accuracy of 0.1 cm (*World Health Organization, 2000*; *Andreacchi et al., 2021*). The averages of the two measurements, both in the waist and in the hip, were used for the analysis. From the results, the waist-to-hip ratio (WHR) was calculated, providing information on the distribution of body fat. In women, values equal to or greater than 0.85 indicate abdominal obesity and values less than 0.85 indicate fat accumulation in the buttocks. In men, the cut-off value for this index is 0.9 (*Nishida, Ko & Kumanyika, 2010*).

### Height

Height was measured using a TANITA HR-001 portable stadiometer. The height of each participant was measured by an investigator in an upright position, with the back to the stadiometer and without shoes, with an accuracy of 0.1 cm (*Norton, 2018*). The average of the three measurements was used for the analysis.

### Body composition analysis

Following the instructions for the Tanita BC-418 MA analyzer used, the weighing platform was placed on a flat surface as possible and levelled to ensure accurate measurement. Each person was also advised to empty their bladder to minimise the risk of error in the analysis of body composition. Subjects were asked to remove excessive clothing (shoes, socks, tights and skin on the soles of the feet were cleaned) and jewellery to increase the precision of the measurement. Measurements were taken in the morning, after at least eight hours of sleep, while fasting (*von Hurst et al., 2016*). BMI was calculated using the formula: body mass(kg)/[height(m)]$^2$. World Health Organization standards were used to classify participants into BMI categories: 18.5–24.99 kg/m$^2$ for normal weight, 25.0–29.99 kg/m$^2$ for overweight, 30.0–34.99 kg/m$^2$ for class I obesity, 35.0–39.99 kg/m$^2$ for class II obesity, ≥40 kg/m$^2$ for class III obesity (*World Health Organization, 2021*). Data from some studies suggest that the optimal range of BMI values for the elderly (≥65 years) should be higher; however, we used cut-off points according to current WHO guidelines (*Jiang et al., 2019*; *Kıskaç et al., 2022*).

### Lipid profile and fasting glucose measurement

Blood samples were taken by finger prick in the morning after an overnight fast. The samples were then analysed using the Cardiocheck PA analyzer, which shows good clinical agreement with the results obtained using a reference laboratory method (*Gao et al., 2016*). TC, HDL, LDL, and TG levels were measured. We used cut-off points for TC (elevated when ≥190 mg/dl), HDL (below standard when ≤45 mg/dl in women; ≤40 mg/dl in men), LDL (elevated when ≥115 mg/dl) and TG (elevated when ≥150 mg/dl) (*Solnica et al., 2020*; *Visseren et al., 2021*). To measure fasting glucose levels, participants were pricked with a finger and their blood glucose levels were measured using a Contour$^{TM}$ Plus One glucose meter. Abnormal fasting glucose was defined as >99 mg/dl (*American Diabetes Association, 2021*; *Araszkiewicz et al., 2023*).
### Blood pressure

Blood pressure measurements were taken according to guidelines (*Unger et al., 2020*). Blood pressure was measured three times using an automatic electronic sphygmomanometer (4200B-E2: Welch Allyn Inc., Aston Abbotts, UK). Cuffs of different widths were used. Systolic blood pressure (SBP) and diastolic blood pressure (DBP) were used for the analysis. The final result was the average of the three measurements. Optimal blood pressure was defined as SBP 120–129 mm Hg and DBP 80–84 mm Hg, elevated blood pressure as SBP 130–139 mm Hg and DBP 85–89, and hypertension was defined as SBP ≥ 140 mmHg and/or DBP ≥ 90 mmHg (*Mancia et al., 2023*). Participants taking medications for hypertension were also classified as hypertensive.

## Statistical analysis

The significance level was taken as $p = 0.05$. Chi-square tests were used to analyze variables expressed at the ordinal or nominal level. For $2 \times 2$ tables, a continuity correction was applied, while for tables larger than $2 \times 2$, if the conditions for using the chi-square test were not met, Fisher's exact test with extension was used. Parametric tests (Student's *T* test) or their nonparametric equivalents (Mann-Whitney *U* test) were used to analyze the quantitative variables presented by group. The choice of tests was based on the distribution of the variables, which was verified by the Shapiro-Wilk test. Variables that did not follow a normal distribution, as determined by the Shapiro-Wilk test, were analyzed using nonparametric tests. The calculations were carried out in the statistical environment R ver. 3.6.0, in the PSPP programme and in MS Office 2019.

## Ethical considerations

The study was carried out according to the Declaration of Helsinki and was approved by the Bioethics Committee of the University of Rzeszów (Resolution No. 2023/07/0046 of 25 June 2023). All participants were informed about the details of the study and were assured anonymity. They gave their informed and voluntary written consent to participate in the study.

## RESULTS

### Characteristics of the group

The study involved 172 people with overweight and obesity (82% female) aged 23 to 85 years who came to a private centre to lose weight. None of the individuals in the final study sample were taking medications for obesity. Furthermore, none of the 172 subjects had undergone bariatric surgery in the past and were not candidates for this type of surgery, nor did they report a history of eating disorders in the questionnaire. The majority of the group had obesity (60.5%), university education (66.3%), were employed (61%), and lived in an urban area (79.1%). Cardiovascular disease (47.7%), thyroid disease (26.2%) and metabolic disease (19.2%) were the most common types of disease in the study group with overweight and obesity. In the total group, 71.5% of subjects had WHR values above the cut-off (*i.e.*, WHR greater than or equal to 0.85 for women and greater than or equal to 0.9 for men), indicating a high prevalence of abdominal obesity among the respondents.

Taking into account the actual results of the physical health parameters, 60.5% of people with overweight and obesity had elevated blood pressure and hypertension, 66.3% had abnormal TC, 55.8% abnormal LDL, 50.6% abnormal TG, and 64.5% abnormal fasting glucose. PA was insufficient in more than half of the study group (54.7%). A total of 22.7% of participants with overweight and obesity had symptoms of depression according to the SDS, and 18.6% met the criteria for FA according to YFAS 2.0. Emotional eating style occurred at a level of 39% and problem eating behaviours occurred at similar levels between groups. The characteristics of the study group are described in the Table 1, taking into account the split between those with FA (individuals who met the YFAS 2.0 criteria for FA) - this includes individuals with mild, moderate or severe levels of FA and NFA (individuals who do not met criteria for FA according to YFAS 2.0).

## Descriptive statistics of the study group

The mean age of all subjects was $M = 59.97$ years ($SD = 11.93$ years), the mean BMI $M = 32.05$ kg/m$^2$ ($SD = 4.84$ kg/m$^2$) and the mean body fat $M = 39.12\%$ ($SD = 6.48$). Descriptive statistics are presented in Table 2 for the study group. Analysis of the Shapiro-Wilk tests for normality of distribution showed that the distribution of most variables was significantly different from normal, necessitating the use of nonparametric tests for such variables.

## Analysis of factors associated with FA

**H1. There are differences in the prevalence of FA according to age**
We found a statistically significant difference in FA scores between age categories $\chi^2(2, N = 172) = 17.23; p = 0.001$ (Table 3). FA was absent in 61.0% of those aged up to 50 years, 82.9% of those aged 51–65 years, and 93.4% of those aged 65 years and over. FA was statistically significantly ($p < 0.05$) more common among in those aged up to 50 years. The older the respondent, the less common the was FA.

**H2. There is a significant difference in the frequency of consumption of different products in subgroups between the subgroup of FA in contrast to NFA**
There were no statistically significant differences ($p > 0.05$) in food group consumption between the FA and NFA groups with overweight and obesity (Table 4). A parametric T-test for independent variables was used for fats, fruits and vegetables and cereal products, comparing the means of the dependent variable between groups. For the other product groups, the non-parametric Mann-Whitney U test was used.

**H3. There is a significant relationship between FA and the number of comorbidities and metabolic parameters such as blood pressure, TC, HDL, LDL, TG, fasting glucose level and anthropometric results**
In the NFA group, the DBP averaged $M = 74.71; SD = 7.65$, while among subjects with FA the mean was higher, at $M = 78.09; SD = 7.45$. The FA group had a statistically significant higher DBP than the NFA subjects, $t(170) = -2.27; p = 0.025$. No significant difference in the number of diseases due to the presence of FA,

$U = 2,223.50$; $p = 0.947$, SBP, $U = 1,986.50$; $p = 0.319$, TC, $U = 2,023.00$; $p = 0.499$, LDL cholesterol $U = 2,145.50$; $p = 0.854$ and HDL cholesterol, $U = 2,025.00$; $p = 0.399$, TG, $U = 2,124.50$; $p = 0.651$ and fasting glucose, $U = 2,040.50$; $p = 0.433$ (Table 5). We also showed that those with FA symptoms had a statistically significant higher BMI, $U = 1,609.00$; $p = 0.013$ and a significantly higher body fat mass, $U = 1,695.00$; $p = 0.032$ (Table 5). The presence of FA did not significantly differentiate the groups considering waist circumference, $U = 1,985.00$; $p = 0.316$ and hips, $U = 1,859.50$; $p = 0.135$, WHR ratio, $U = 2,119.50$; $p = 0.636$ and amount of visceral tissue, $U = 1,993.50$; $p = 0.330$.

## The relationship between FA and depressive symptoms

**H4. There is a significant relationship between BMI, eating behaviours, PA and SB, and symptoms of FA and depression**

### BMI

The majority of individuals, regardless of weight, did not have FA and depressive symptoms. FA alone was shown to be statistically significantly more common in those with class II or III obesity than in the others $\chi^2(6, N = 172) = 13.19$; $p = 0.040$ (Table 6). However, there were no significant differences in depressive symptoms between the BMI groups.

### Eating behaviours

Individuals with uncontrolled (66.1%), emotional (71.6%) and restrictive (58.7%) eating styles were more likely to be free of symptoms of FA and depression. The result was not statistically significant, $\chi^2(6, N = 172) = 4.75$; $p = 0.576$. No significant relationship ($p > 0.05$) was observed between problematic eating behaviour and FA and depression symptoms (Table 6).

### PA and SB

The results of the chi-square analysis showed a statistically significant difference in the distribution of the occurrence of FA and depressive symptoms according to the level of SB, $\chi^2(6, N = 172) = 21.61$; $p = 0.001$. Subjects with lower SB were significantly less likely to have FA and depressive symptoms compared to those with higher SB. In contrast, those with SB between 301–450 and above 450 were significantly more likely to have symptoms of depression and FA, respectively (Table 6). The relationship between PA levels and the presence of FA and/or depressive symptoms was not statistically significant $\chi^2(3, N = 172) = 5.26$; $p = 0.154$. Among those with insufficient PA and those with sufficient activity, symptoms of FA and depressive symptoms were equally likely to be absent (66.0% and 66.7% respectively).

## DISCUSSION

The increasing prevalence of excess body weight and the ineffectiveness of traditional weight loss approaches have prompted a quest to identify factors contributing to obesity

**Table 1  Characteristics of the study group.**

| Variable | | Food addiction | | | | | |
|---|---|---|---|---|---|---|---|
| | | NFA | | FA | | Total | |
| | | N | % | N | % | N | % |
| Gender | Female | 116 | 82.9% | 25 | 78.1% | 141 | 82.0% |
| | Male | 24 | 17.1% | 7 | 21.9% | 31 | 18.0% |
| Age range | Up to 50 years of age | 25 | 17.9% | 16 | 50.0% | 41 | 23.8% |
| | 51–65 years of age | 58 | 41.4% | 12 | 37.5% | 70 | 40.7% |
| | Over 65 years of age | 57 | 40.7% | 4 | 12.5% | 61 | 35.5% |
| Education | Primary education | 1 | 0.7% | 0 | 0% | 1 | 0.6% |
| | Lower secondary education | 3 | 2.1% | 1 | 3.1% | 4 | 2.3% |
| | Vocational education | 9 | 6.4% | 2 | 6.3% | 11 | 6.4% |
| | Secondary education | 34 | 24.3% | 8 | 25.0% | 42 | 24.4% |
| | Higher education | 93 | 66.4% | 21 | 65.6% | 114 | 66.3% |
| Place of residence | Village | 29 | 20.7% | 7 | 21.9% | 36 | 20.9% |
| | City | 111 | 79.3% | 25 | 78.1% | 136 | 79.1% |
| Comorbidities | Metabolic diseases | 28 | 20.0% | 5 | 15.6% | 33 | 19.2% |
| | Cardiovascular diseases | 67 | 47.9% | 15 | 46.9% | 82 | 47.7% |
| | Thyroid diseases | 34 | 24.3% | 11 | 34.4% | 45 | 26.2% |
| | Respiratory diseases | 18 | 12.9% | 2 | 6.3% | 20 | 11.6% |
| | Diseases of the digestive system | 22 | 15.7% | 1 | 3.1% | 23 | 13.4% |
| | BED | 2 | 1.4% | 3 | 9.4% | 5 | 2.9% |
| | Cancer diseases | 2 | 1.4% | 1 | 3.1% | 3 | 1.7% |
| | Osteoarticular diseases | 7 | 5.0% | 1 | 3.1% | 8 | 4.7% |
| | Not applicable | 35 | 25.0% | 7 | 21.9% | 42 | 24.4% |
| Professional activity | Retirement/Pension | 62 | 44.3% | 5 | 15.6% | 67 | 39.0% |
| | Occasional workers | 8 | 5.7% | 3 | 9.4% | 11 | 6.4% |
| | Permanent employment | 70 | 50.0% | 24 | 75.0% | 94 | 54.7% |
| Employment form | Not working | 62 | 44.3% | 5 | 15.6% | 67 | 39.0% |
| | Working | 78 | 55.7% | 27 | 84.4% | 105 | 61.0% |
| BMI classification (kg/m$^2$) | Overweight | 60 | 42.9% | 8 | 25.0% | 68 | 39.5% |
| | Class I obesity | 52 | 37.1% | 12 | 37.5% | 64 | 37.2% |
| | Class II obesity | 17 | 12.1% | 6 | 18.8% | 23 | 13.4% |
| | Class III obesity | 11 | 7.9% | 6 | 18.8% | 17 | 9.9% |
| Blood pressure (mmHg) | Optimal | 56 | 40.0% | 12 | 37.5% | 68 | 39.5% |
| | Elevated | 9 | 6.4% | 4 | 12.5% | 13 | 7.6% |
| | Hypertension | 75 | 53.6% | 16 | 50.0% | 91 | 52.9% |
| TC (mg/dl) | Acceptable | 44 | 31.4% | 14 | 43.8% | 58 | 33.7% |
| | Abnormal | 96 | 68.6% | 18 | 56.3% | 114 | 66.3% |
| LDL (mg/dl) | Acceptable | 61 | 43.6% | 15 | 46.9% | 76 | 44.2% |
| | Abnormal | 79 | 56.4% | 17 | 53.1% | 96 | 55.8% |
| HDL (mg/dl) | Acceptable | 86 | 61.4% | 21 | 65.6% | 107 | 62.2% |
| | Abnormal | 54 | 38.6% | 11 | 34.4% | 65 | 37.8% |

| Variable | | Food addiction | | | | | |
|---|---|---|---|---|---|---|---|
| | | NFA | | FA | | Total | |
| | | N | % | N | % | N | % |
| TG (mg/dl) | Acceptable | 66 | 47.1% | 19 | 59.4% | 85 | 49.4% |
| | Abnormal | 74 | 52.9% | 13 | 40.6% | 87 | 50.6% |
| Fasting blood glucose (mg/dl) | Acceptable | 50 | 35.7% | 11 | 34.4% | 61 | 35.5% |
| | Abnormal | 90 | 64.3% | 21 | 65.6% | 111 | 64.5% |
| Level of PA | Insufficient | 80 | 57.1% | 14 | 43.8% | 94 | 54.7% |
| | Sufficient | 60 | 42.9% | 18 | 56.3% | 78 | 45.3% |
| Smoking | Yes | 17 | 12.1% | 10 | 31.3% | 27 | 15.7% |
| | No | 123 | 87.9% | 22 | 68.8% | 145 | 84.3% |
| Alcohol consumption | Several times a month or less frequently | 114 | 81.4% | 21 | 65.6% | 135 | 78.5% |
| | Several times a week and more often | 26 | 18.6% | 11 | 34.4% | 37 | 21.5% |
| Workday sleep time (hours/day) | six or less | 55 | 39.3% | 14 | 43.8% | 69 | 40.1% |
| | 7–8 | 72 | 51.4% | 15 | 46.9% | 87 | 50.6% |
| | nine and more | 13 | 9.3% | 3 | 9.4% | 16 | 9.3% |
| Weekend sleep time (hours/day) | six or less | 29 | 20.7% | 6 | 18.8% | 35 | 20.3% |
| | 7–8 | 82 | 58.6% | 15 | 46.9% | 97 | 56.4% |
| | nine and more | 29 | 20.7% | 11 | 34.4% | 40 | 23.3% |
| Depressive symptoms | Lack of symptoms | 114 | 81.4% | 19 | 59.4% | 133 | 77.3% |
| | Minor to mild depression | 23 | 16.4% | 11 | 34.4% | 34 | 19.8% |
| | Moderate to significant depression | 3 | 2.1% | 2 | 6.3% | 5 | 2.9% |
| Severity of FA | Lack of symptoms | 140 | 100.0% | 0 | 0% | 140 | 81.4% |
| | Mild | 0 | 0% | 10 | 31.3% | 10 | 5.8% |
| | Moderate | 0 | 0% | 8 | 25.0% | 8 | 4.7% |
| | Severe | 0 | 0% | 14 | 43.8% | 14 | 8.1% |
| Eating behaviours | Uncontrolled eating | 47 | 33.6% | 12 | 37.5% | 59 | 34.3% |
| | Emotional eating | 58 | 41.4% | 9 | 28.1% | 67 | 39.0% |
| | Cognitive restraint | 35 | 25.0% | 11 | 34.4% | 46 | 26.7% |
| Cooccurrence of FA and depression symptoms | Absence of symptoms of FA and depression | 114 | 81.4% | 0 | 0% | 114 | 66.3% |
| | Depressive symptoms | 26 | 18.6% | 0 | 0% | 26 | 15.1% |
| | FA | 0 | 0% | 19 | 59.4% | 19 | 11.0% |
| | Depressive symptoms and FA | 0 | 0% | 13 | 40.6% | 13 | 7.6% |

**Note:**
*N*, abundance; BED, Binge Eating Disorder; BMI, Body Mass Index; TC, total cholesterol; LDL, low-density lipoprotein; HDL, high-density lipoprotein; TG, triglycerides; FA, Food Addiction; PA, physical activity.

development. With overweight and obesity posing significant challenges to both physical and mental well-being on a global scale, there is a pressing need for tailored treatment strategies. Thus, this study aimed to evaluate the physical and mental health profiles of

| Variable | FA | N | M | SD | Min | Maks | p | |
|---|---|---|---|---|---|---|---|---|
| Age | NFA | 140 | 61.52 | 11.54 | 28.00 | 85.00 | <0.001 | |
| | FA | 32 | 53.16 | 11.38 | 23.00 | 74.00 | 0.725 | |
| | Total | 172 | 59.97 | 11.93 | 23.00 | 85.00 | 0.001 | |
| Body weight (kg) | NFA | 140 | 84.70 | 15.25 | 56.90 | 135.70 | <0.001 | *** |
| | FA | 32 | 93.42 | 18.53 | 62.50 | 136.00 | 0.243 | |
| | Total | 172 | 86.32 | 16.21 | 56.90 | 136.00 | <0.001 | *** |
| Height (cm) | NFA | 140 | 163.58 | 8.05 | 142.00 | 184.00 | 0.010 | * |
| | FA | 32 | 164.94 | 7.35 | 155.00 | 183.00 | 0.064 | |
| | Total | 172 | 163.83 | 7.92 | 142.00 | 184.00 | 0.002 | ** |
| BMI (kg/m$^2$) | NFA | 140 | 31.56 | 4.52 | 25.10 | 47.60 | <0.001 | *** |
| | FA | 32 | 34.18 | 5.64 | 26.00 | 47.30 | 0.218 | |
| | Total | 172 | 32.05 | 4.84 | 25.10 | 47.60 | <0.001 | *** |
| Fat mass (%) | NFA | 140 | 38.66 | 6.35 | 18.80 | 53.10 | <0.001 | *** |
| | FA | 32 | 41.11 | 6.74 | 22.80 | 53.90 | 0.197 | |
| | Total | 172 | 39.12 | 6.48 | 18.80 | 53.90 | <0.001 | *** |
| Fat-free mass (kg) | NFA | 140 | 51.46 | 10.86 | 16.40 | 85.30 | <0.001 | *** |
| | FA | 32 | 54.71 | 11.33 | 38.60 | 82.80 | 0.002 | ** |
| | Total | 172 | 52.07 | 10.99 | 16.40 | 85.30 | <0.001 | *** |
| Visceral fat | NFA | 140 | 11.91 | 3.72 | 5.00 | 27.00 | <0.001 | *** |
| | FA | 32 | 13.09 | 4.99 | 7.00 | 30.00 | <0.001 | *** |
| | Total | 172 | 12.13 | 4.00 | 5.00 | 30.00 | <0.001 | *** |
| Waist circumference (cm) | NFA | 140 | 100.18 | 12.41 | 81.00 | 140.00 | <0.001 | *** |
| | FA | 32 | 103.28 | 14.14 | 80.00 | 137.00 | 0.182 | |
| | Total | 172 | 100.76 | 12.76 | 80.00 | 140.00 | <0.001 | *** |
| Hip circumference (cm) | NFA | 140 | 109.70 | 9.77 | 89.00 | 149.00 | 0.012 | * |
| | FA | 32 | 113.41 | 12.58 | 90.00 | 139.00 | 0.750 | |
| | Total | 172 | 110.39 | 10.41 | 89.00 | 149.00 | 0.010 | * |
| WHR | NFA | 140 | 0.92 | 0.13 | 0.71 | 1.54 | <0.001 | *** |
| | FA | 32 | 0.92 | 0.13 | 0.76 | 1.27 | 0.001 | ** |
| | Total | 172 | 0.92 | 0.13 | 0.71 | 1.54 | <0.001 | *** |
| SBP (mmHg) | NFA | 140 | 123.20 | 15.35 | 94.00 | 175.00 | 0.002 | ** |
| | FA | 32 | 125.22 | 13.61 | 100.00 | 153.00 | 0.774 | |
| | Total | 172 | 123.58 | 15.02 | 94.00 | 175.00 | 0.003 | ** |
| DBP (mmHg) | NFA | 140 | 74.71 | 7.65 | 54.00 | 93.00 | 0.689 | |
| | FA | 32 | 78.09 | 7.45 | 64.00 | 95.00 | 0.883 | |
| | Total | 172 | 75.34 | 7.71 | 54.00 | 95.00 | 0.874 | |
| TC (mg/dl) | NFA | 137 | 195.79 | 42.25 | 101.00 | 329.00 | 0.529 | |
| | FA | 32 | 196.06 | 56.25 | 102.00 | 359.00 | 0.007 | ** |
| | Total | 169 | 195.84 | 45.04 | 101.00 | 359.00 | 0.004 | ** |
| LDL (mg/dl) | NFA | 137 | 108.76 | 36.34 | 24.00 | 228.00 | 0.132 | |
| | FA | 32 | 110.44 | 47.43 | 37.00 | 248.00 | 0.005 | ** |
| | Total | 169 | 109.08 | 38.53 | 24.00 | 248.00 | 0.001 | ** |

Table 2 Descriptive statistics of the study group.

| Variable | FA | N | M | SD | Min | Maks | p | |
|---|---|---|---|---|---|---|---|---|
| HDL (mg/dl) | NFA | 140 | 57.33 | 15.46 | 27.00 | 104.00 | <0.001 | *** |
| | FA | 32 | 54.94 | 15.64 | 28.00 | 88.00 | 0.352 | |
| | Total | 172 | 56.88 | 15.48 | 27.00 | 104.00 | <0.001 | *** |
| TG (mg/dl) | NFA | 140 | 147.14 | 63.23 | 50.00 | 385.00 | <0.001 | *** |
| | FA | 32 | 144.13 | 63.93 | 63.00 | 328.00 | 0.001 | ** |
| | Total | 172 | 146.58 | 63.18 | 50.00 | 385.00 | <0.001 | *** |
| Fasting blood glucose (mg/dl) | NFA | 140 | 108.15 | 17.26 | 82.00 | 183.00 | <0.001 | *** |
| | FA | 32 | 103.84 | 13.49 | 81.00 | 138.00 | 0.591 | |
| | Total | 172 | 107.35 | 16.67 | 81.00 | 183.00 | <0.001 | *** |
| SB (min/day) | NFA | 140 | 339.86 | 156.09 | 60.00 | 720.00 | <0.001 | *** |
| | FA | 32 | 452.97 | 152.06 | 120.00 | 720.00 | 0.012 | * |
| | Total | 172 | 360.90 | 161.07 | 60.00 | 720.00 | <0.001 | *** |
| Total activity (MET-min/week) | NFA | 140 | 1,866.71 | 3,362.22 | 0.00 | 20,040.00 | <0.001 | *** |
| | FA | 32 | 2,132.50 | 3,421.42 | 0.00 | 14,400.00 | <0.001 | *** |
| | Total | 172 | 1,916.16 | 3,364.82 | 0.00 | 20,040.00 | <0.001 | *** |
| Self-reported depressive symptoms | NFA | 140 | 38.91 | 9.51 | 20.00 | 68.00 | 0.070 | |
| | FA | 32 | 43.66 | 10.95 | 26.00 | 64.00 | 0.220 | |
| | Total | 172 | 39.79 | 9.93 | 20.00 | 68.00 | 0.024 | * |
| Uncontrolled eating | NFA | 140 | 56.23 | 13.23 | 18.50 | 85.20 | 0.025 | * |
| | FA | 32 | 52.09 | 15.68 | 14.80 | 77.80 | 0.479 | |
| | Total | 172 | 55.46 | 13.76 | 14.80 | 85.20 | 0.004 | ** |
| Emotional eating | NFA | 140 | 51.59 | 29.40 | 0.00 | 100.00 | <0.001 | *** |
| | FA | 32 | 35.76 | 27.04 | 0.00 | 100.00 | 0.048 | * |
| | Total | 172 | 48.64 | 29.55 | 0.00 | 100.00 | <0.001 | *** |
| Cognitive restraint | NFA | 140 | 53.14 | 10.03 | 27.80 | 77.80 | 0.001 | ** |
| | FA | 32 | 50.51 | 12.26 | 22.20 | 72.20 | 0.195 | |
| | Total | 172 | 52.65 | 10.50 | 22.20 | 77.80 | 0.001 | ** |

**Note:**
*N*, abundance; *M*, mean; *SD*, standard deviation; *Min*, minimum; *Maks*, maksimum; FA, food addiction; BMI, Body Mass Index; WHR, waist-to-hip ratio; SBP, systolic blood pressure; DBP, diastolic blood pressure; TC, total cholesterol; LDL, low-density lipoprotein; HDL, high-density lipoprotein; TG, triglycerides; SB, sedentary behaviour. $^{*} p < 0.05$; $^{**} p < 0.01$; $^{***} p < 0.001$

**Table 3 Prevalence of FA by age.**

| | | | Age range | | | Test result |
|---|---|---|---|---|---|---|
| | | | Up to 50 years | 51–65 years of age | More than 65 years | |
| **FA** | NFA | N | 25 | 58 | 57 | $\chi^2 = 17.234$ |
| | | % | 61.0% | 82.9% | 93.4% | $df = 2$ |
| | FA | N | 16 | 12 | 4 | $p = 0.001$ |
| | | % | 39.0% | 17.1% | 6.6% | |
| Total | | N | 41 | 70 | 61 | |
| | | % | 100.0% | 100.0% | 100.0% | |

**Note:**
$\chi^2$, test statistics; *df*, degrees of freedom; *N*, abundance; *p*, statistical significance; FA, food addiction.

**Table 4 Differences in frequency of consumption of product groups by FA.**

|  |  | t/U | df | p | M | SD | Min | Maks |
|---|---|---|---|---|---|---|---|---|
| FA | Sweets and snacks | 1,954.00 |  | 0.261 |  |  |  |  |
|  | NFA |  |  |  | 3.04 | 0.84 | 1.43 | 4.86 |
|  | FA |  |  |  | 3.23 | 0.83 | 1.43 | 4.71 |
|  | Dairy products and eggs | 2,011.50 |  | 0.368 |  |  |  |  |
|  | NFA |  |  |  | 3.25 | 0.74 | 1.67 | 5.00 |
|  | FA |  |  |  | 3.35 | 0.76 | 1.83 | 4.67 |
|  | Cereal products | 2,064.00 |  | 0.487 |  |  |  |  |
|  | NFA |  |  |  | 3.05 | 0.71 | 1.00 | 4.60 |
|  | FA |  |  |  | 3.10 | 0.82 | 1.00 | 4.80 |
|  | Fats | 0.16 | 41 | 0.873 |  |  |  |  |
|  | NFA |  |  |  | 2.93 | 0.59 | 1.00 | 4.50 |
|  | FA |  |  |  | 2.91 | 0.72 | 1.83 | 4.50 |
|  | Fruits | −0.17 | 170 | 0.862 |  |  |  |  |
|  | NFA |  |  |  | 3.10 | 0.60 | 1.36 | 4.55 |
|  | FA |  |  |  | 3.12 | 0.46 | 2.18 | 3.91 |
|  | Vegetables and grains | 0.38 | 170 | 0.706 |  |  |  |  |
|  | NFA |  |  |  | 3.39 | 0.53 | 1.58 | 4.75 |
|  | FA |  |  |  | 3.35 | 0.52 | 2.17 | 4.75 |
|  | Meat and fish products | 2,044.50 |  | 0.441 |  |  |  |  |
|  | NFA |  |  |  | 2.58 | 0.51 | 1.00 | 4.00 |
|  | FA |  |  |  | 2.67 | 0.58 | 1.25 | 4.00 |
|  | Beverages | 1,798.50 |  | 0.082 |  |  |  |  |
|  | NFA |  |  |  | 2.15 | 0.80 | 1.00 | 4.29 |
|  | FA |  |  |  | 2.42 | 0.90 | 1.00 | 4.29 |

Note:
*t*, test statistic; *U*, test statistic; *df*, degrees of freedom; *p*, statistical significance; *M*, mean; *SD*, standard deviation; *Min*, minimum; *Maks*, maksimum; FA, food addiction.

**Table 5 Physical health parameters related to FA.**

Comorbidities and metabolic parameters:

|  |  | t/U | p | Min | Maks |
|---|---|---|---|---|---|
| FA | Number of diseases | 2,223.50 | 0.947 |  |  |
|  | NFA |  |  | 0.00 | 4.00 |
|  | FA |  |  | 0.00 | 3.00 |
|  | SBP | 1,986.50 | 0.319 |  |  |
|  | NFA |  |  | 94.00 | 175.00 |
|  | FA |  |  | 100.00 | 153.00 |
|  | DBP | −2.27 | 0.025 |  |  |
|  | NFA |  |  | 54.00 | 93.00 |
|  | FA |  |  | 64.00 | 95.00 |

*Comorbidities and metabolic parameters:*

|  |  | t/U | p | Min | Maks |
|---|---|---|---|---|---|
|  | **TC** | 2,023.00 | 0.499 |  |  |
|  | NFA |  |  | 101.00 | 329.00 |
|  | FA |  |  | 102.00 | 359.00 |
|  | **LDL** | 2,145.50 | 0.854 |  |  |
|  | NFA |  |  | 24.00 | 228.00 |
|  | FA |  |  | 37.00 | 248.00 |
|  | **HDL** | 2,025.00 | 0.399 |  |  |
|  | NFA |  |  | 27.00 | 104.00 |
|  | FA |  |  | 28.00 | 88.00 |
|  | **TG** | 2,124.50 | 0.651 |  |  |
|  | NFA |  |  | 50.00 | 385.00 |
|  | FA |  |  | 63.00 | 328.00 |
|  | **Fasting glucose** | 2,040.50 | 0.433 |  |  |
|  | NFA |  |  | 82.00 | 183.00 |
|  | FA |  |  | 81.00 | 138.00 |

*Anthropometric parameters:*

|  |  | U | p | Min | Maks |
|---|---|---|---|---|---|
| **FA** | **BMI** | 1,609.00 | 0.013 |  |  |
|  | NFA |  |  | 25.10 | 47.60 |
|  | FA |  |  | 26.00 | 47.30 |
|  | **Waist circumference** | 1,985.00 | 0.316 |  |  |
|  | NFA |  |  | 81.00 | 140.00 |
|  | FA |  |  | 80.00 | 137.00 |
|  | **Hip circumference** | 1,859.50 | 0.135 |  |  |
|  | NFA |  |  | 89.00 | 149.00 |
|  | FA |  |  | 90.00 | 139.00 |
|  | **WHR** | 2,119.50 | 0.636 |  |  |
|  | NFA |  |  | 0.71 | 1.54 |
|  | FA |  |  | 0.76 | 1.27 |
|  | **Fat mass** | 1,695.00 | 0.032 |  |  |
|  | NFA |  |  | 18.80 | 53.10 |
|  | FA |  |  | 22.80 | 53.90 |
|  | **Visceral fat** | 1,993.50 | 0.330 |  |  |
|  | NFA |  |  | 5.00 | 27.00 |
|  | FA |  |  | 7.00 | 30.00 |

**Note:**

U, test statistics; p, statistical significance; Min, minimum score; Maks, maximum score; FA, food addiction.

adults with overweight and obesity undergoing weight loss efforts. This evaluation was based on precise measurements of metabolic and anthropometric parameters, comprehensive questionnaires, and a lifestyle analysis.

**Table 6 BMI, eating behaviours, SB, PA and symptoms of FA and depression.**

| | | | BMI | | | Test result |
|---|---|---|---|---|---|---|
| | | | Overweight | Class I obesity | Class II/III obesity | |
| **Depressive symptoms and FA** | Absence of FA and depressive symptoms | N | 48 | 44 | 22 | |
| | | % | 70.6% | 68.8% | 55.0% | |
| | Depressive symptoms | N | 12 | 8 | 6 | |
| | | % | 17.6% | 12.5% | 15.0% | $\chi^2 = 13.186$ |
| | FA | N | 2 | 7 | 10 | $df = 6$ |
| | | % | 2.9% | 10.9% | 25.0% | $p = 0.040$ |
| | Depressive symptoms and FA | N | 6 | 5 | 2 | |
| | | % | 8.8% | 7.8% | 5.0% | |
| Total | | N | 68 | 64 | 40 | |
| | | % | 100.0% | 100.0% | 100.0% | |

| | | | Eating behaviours | | | Test result |
|---|---|---|---|---|---|---|
| | | | Uncontrolled eating | Emotional eating | Restrictive eating | |
| **Depressive symptoms and FA** | Absence of FA and depressive symptoms | N | 39 | 48 | 27 | |
| | | % | 66.1% | 71.6% | 58.7% | |
| | Depressive symptoms | N | 8 | 10 | 8 | |
| | | % | 13.6% | 14.9% | 17.4% | $\chi^2 = 4.753$ |
| | FA | N | 7 | 7 | 5 | $df = 6$ |
| | | % | 11.9% | 10.4% | 10.9% | $p = 0.576$ |
| | Depressive symptoms and FA | N | 5 | 2 | 6 | |
| | | % | 8.5% | 3.0% | 13.0% | |
| Total | | N | 59 | 67 | 46 | |
| | | % | 100% | 100% | 100% | |

| | | | SB | | | Test result |
|---|---|---|---|---|---|---|
| | | | Up to 300 min/day | 301–450 min/day | More than 450 min/day | |
| **Depressive symptoms and FA** | Absence of FA and depressive symptoms | N | 64 | 18 | 32 | |
| | | % | 81.0% | 54.5% | 53.3% | |
| | Depressive symptoms | N | 8 | 9 | 9 | |
| | | % | 10.1% | 27.3% | 15.0% | $\chi^2 = 21.608$ |
| | FA | N | 4 | 2 | 13 | $df = 6$ |
| | | % | 5.1% | 6.1% | 21.7% | $p = 0.001$ |
| | Depressive symptoms and FA | N | 3 | 4 | 6 | |
| | | % | 3.8% | 12.1% | 10.0% | |
| Total | | N | 79 | 33 | 60 | |
| | | % | 100.0% | 100.0% | 100.0% | |

| | | | Level of PA | | Test result |
|---|---|---|---|---|---|
| | | | Insufficient | Sufficient | |
| **Depressive symptoms and FA** | Absence of FA and depressive symptoms | N | 62 | 52 | |
| | | % | 66.0% | 66.7% | |

| Table 6 (continued) | | | Level of PA | | Test result |
|---|---|---|---|---|---|
| | | | Insufficient | Sufficient | |
| Depressive symptoms | | N | 18 | 8 | |
| | | % | 19.1% | 10.3% | $\chi^2 = 5,256$ |
| FA | | N | 10 | 9 | $df = 3$ |
| | | % | 10.6% | 11.5% | $p = 0.154$ |
| Depressive symptoms and FA | | N | 4 | 9 | |
| | | % | 4.3% | 11.5% | |
| Total | | N | 94 | 78 | |
| | | % | 100.0% | 100.0% | |

**Note:**
$\chi^2$, test statistics; *df*, degrees of freedom; *N*, abundance; *p*, statistical significance; FA, Food Addiction; BMI, Body Mass Index; SB, sedentary behaviour; PA, physical activity.

## There are differences in the prevalence of FA according to age

In our study, the prevalence of FA was 18.6% among people with overweight and obesity. *Pape et al. (2021)* reported a slightly lower level of FA in participants enrolled in a weight loss programme, where the prevalence of FA was 15%. *Ivezaj, White & Grilo (2016)* and *Imperatori et al. (2017)* found a higher prevalence in subjects with overweight and obesity, with 26.7% and 22.9% respectively meeting FA criteria, but BED was more common in these groups. Therefore, the lower prevalence in our study may be due to the fact that none of the participants were candidates for bariatric surgery and had a history of eating disorders. There were also no cases of eating disorders among the study participants, except for a low prevalence of BED (2.9%). Significantly lower results were found in a study of veterans with overweight/obesity, where 10% of the sample reached the diagnostic level of FA, and in a study of individuals with obesity trying to lose weight, FA occurred in 6.7% of subjects (*Chao et al., 2017*; *Masheb et al., 2018*). In a meta-analysis, *Pursey et al. (2014)* showed that approximately one in four people with overweight/obesity has FA, and that adults over 35 years of age were more likely to meet the criteria for FA than those under 35 years of age (22.2% *vs.* 17%), and the prevalence of FA was twice as high in women as in men. In our study, FA was statistically significantly more common in those over 50 years of age, and the older the subjects were, the less common FA was, which is consistent with most previous studies (*Pursey et al., 2014*; *Flint et al., 2014*).

## There is no significant difference in the frequency of consumption of different products in subgroups between the subgroup of FA in contrast to NFA

*Reche-García et al. (2022)* highlighted the impact of addictive behaviours on the eating habits of people with overweight or obesity. In studies of people with overweight and obesity in which dietary patterns were measured in detail, people with FA consumed significantly more energy-rich and processed foods and thus had higher intakes of energy, carbohydrate, total fat and protein, saturated fat, and sugar intakes than people without FA

(*Pedram & Sun, 2014*; *Davis et al., 2014*; *Moghaddam et al., 2019*). This was also shown in a large epidemiological study among nurses, where FA was positively associated with the consumption of highly processed and palatable foods, i.e., fast food, snacks/desserts, chocolate bars, while it was inversely or not associated with some sugary foods, refined cereals, and sugar-sweetened beverages (*Lemeshow et al., 2018*). In our study, there were no significant differences in the frequency of food consumption of the food groups between people with overweight and obesity with FA and those with NFA. *Kiyici et al. (2020)* also found no differences in the dietary habits of people with obesity with and without FA. This may be because the respondents are not fully aware of their eating habits or are reluctant to admit them. There is a strong emotional and psychological aspect to FA and eating habits that can affect the honesty of responses (*Reche-García et al., 2022*). Importantly, FFQs cannot cover the full range of foods consumed, including ultra-processed foods, which *Gearhardt & DiFeliceantonio (2023)* say are the predominant source of addictive foods in the modern world.

## There is no significant relationship between FA and the number of comorbidities and metabolic parameters such as SBP, TC, HDL, LDL, TG, fasting glucose level, waist and hip circumferences, WHR, and visceral fat

### Comorbidities and metabolic parameters

It is not yet clear whether FA is related or even contributes to obesity-related comorbidities. In patients with obesity with a BMI $\geq$ 35 kg/m$^2$ referred for bariatric surgery, FA was not associated with obesity-related complications such as cardiovascular disease, including hypertension, obstructive sleep apnea syndrome (OSAS), and type 2 diabetes (*Som et al., 2018*). In general, FA can be considered a potential contributor to obesity, but not to its complications, which are also due to metabolic, environmental, or genetic factors. In a study by *Kiyici et al. (2020)* in patients with obesity, age- and sex-adjusted mean fasting glucose was lower in patients with FA, but serum insulin levels and the lipid parameter index were also comparable to the group without FA. In our study, the presence of FA did not significantly differentiate the number of comorbidities, SBP, TC, LDL and HDL cholesterol, TG, and fasting glucose. However, the FA group had a statistically significantly higher DBP than those with NFA. The *Horsager et al. (2023)* study showed that there is a strong and intuitively significant positive association between FA and type 2 diabetes. The strength of the positive association between FA and type 2 diabetes was somewhat attenuated after adjustment for BMI, suggesting that BMI/obesity may be the causal factor in the association, but not the only one.

In this case, although excess body weight may be a key factor affecting some physical health parameters, such as blood pressure, the presence of FA can also have a significant impact on these parameters. In conclusion, our findings support the view that people with overweight and obesity who participate in various weight loss programmes may experience addictive eating behaviours and depressive symptoms. These disorders can contribute to weight gain, make weight loss more difficult, or make it more difficult to maintain

long-term weight loss. Our results confirm that individualised and multidisciplinary efforts to support overweight people trying to lose weight may be warranted.

### Anthropometric results

In previous studies, the severity of FA has been associated with an increase in BMI and body fat (*Gearhardt, Boswell & White, 2014*; *Pape et al., 2021*). This is consistent with our findings. However, there were no significant differences in waist and hip circumferences, WHR, and visceral fat content in our study.

## There is a significant relationship between BMI and SB and symptoms of FA and depression but not with eating behaviours and PA

### Symptoms of depression

In a systematic review and meta-analysis, *Burrows et al. (2018)* found a significant positive association between FA and mental health symptoms, including depression. In our study, 22.7% of people with overweight and obesity had depressive symptoms. *Kiyici et al. (2020)* showed that of 224 patients with obesity seeking weight loss treatment, 32.1% met the criteria for FA and the prevalence of depression was significantly higher (61.1%) in patients with FA compared to those without. Interestingly, the study by *Usta & Pehlivan (2021)* found that depressive symptoms mediated the association between FA symptoms and BMI in people with obesity. The result of the meta-analysis by *de Sousa Fernandes et al. (2020)* showed that, among individuals with overweight/obesity, FA was present in those with the highest BMI levels. Furthermore, the meta-analysis identified high depression scores in individuals with FA.

### BMI

In the study by *Bartschi & Greenwood (2023)*, FA symptoms fully mediated the association between the severity of depression symptoms and BMI. In our study there were no significant differences in depressive symptoms between the BMI groups but FA was shown to be statistically significantly more common in those with class II or III obesity. People with obesity showing a worse weight loss response to treatment and greater weight gain after undergoing bariatric surgery obtain higher level of FA (*Burmeister et al., 2013*; *Clark & Saules, 2013*). Therefore, early detection of FA and prevention of obesity development seems to be an important issue and weight loss treatments should consider the role of FA as a psychological factor underlying difficult weight management situations.

### Eating behaviours

Some of the data suggest an association between FA and food-related variables, such as eating behaviours, which may be related to this excessive food intake (*Pepino et al., 2014*). The results of previous studies have shown that FA was positively associated with emotional and uncontrolled eating (*Chao et al., 2017*; *Escrivá-Martínez et al., 2023*; *Schankweiler et al., 2023*; *Wattick et al., 2023*). Emotional eating has been associated with mental health (depression, anxiety), overweight and/or obesity and unhealthy eating patterns (*Dakanalis et al., 2023*). The emotional eating style was observed at a level of 39%

in our study group and we found no significant statistical differences in the eating styles between FA and NFA. Both those with uncontrolled, emotional, and restrictive eating behaviours were more likely to be free of FA and depressive symptoms, although the differences were small and not statistically significant.

### SB and PA

More than one-third (36.2%) of adults in Europe were physically inactive (*Nikitara et al., 2021*). In the 65–75 age group, approximately 55–83% of women and 47–74% of men do not meet WHO recommendations for PA (*Lübs et al., 2018*; *Bull et al., 2020*). In a cross-sectional study of relatively healthy adults aged 70 and over recruited from the community in five European countries, at baseline, 62.2% reported meeting PA recommendations and 37.1% classified as sedentary (reported to spend ≥5.5 h/day of SB) (*Mattle et al., 2022*). In our study, 54.7% of all participants with excess body weight had insufficient levels of PA, which meant they did not meet WHO recommendations (*World Health Organization, 2020*). Unlike the findings of other researchers, we did not obtain statistical significance when comparing PA of overweight individuals with FA and NFA. However, depressive symptoms were significantly more common among those who were sedentary for 301–450 min/day, and FA symptoms were significantly more common among those who were sedentary for more than 450 min/day. *Romero-Blanco et al. (2021)* obtained similar results, showing that FA was associated with a SB, but not with PA. Previous research also suggests that people with FA spend less time doing moderate to vigorous physical activity (MVPA) and more time sedentary (*Li et al., 2018*). It is clear that low levels of PA and SB are associated with appetite regulation, snacking, and a greater preference and desire for energy-dense foods (*Beaulieu, Oustric & Finlayson, 2020*; *Beltrán-Carrillo et al., 2022*). The WHO stresses the importance of reducing SB in addition to meeting the recommendations of PA (*Bull et al., 2020*). *Park et al. (2020)* emphasises that SB have wide-ranging adverse impacts on the human body including increased all-cause mortality, cardiovascular disease mortality, cancer risk, and risks of metabolic disorders such as diabetes mellitus, hypertension, and dyslipidemia, musculoskeletal disorders, depression and cognitive impairment.

## Limitations

In addition to anthropometric measurements, blood parameters and blood pressure, some of our conclusions are based on the results of self-report questionnaires, and different results can be observed for behavioural measures, as shown by *Loeber et al. (2018)*. Interpreting our results is challenging due to the cross-sectional nature of the study, which does not allow causality to be fully established. Long-term, longitudinal studies are needed to provide a more accurate analysis and verify our observations.

The FFQ-6 questionnaire we used in the study does not include fast food products, which is a major limitation and makes it impossible to determine the consumption of these products in the context of FA assessment. However, these limitations provide valuable information to consider the development of new dietary assessment methods to evaluate

intakes associated with FA or UPF foods according to, for example, the NOVA classification (*Monteiro et al., 2019*).

Another limitation may be the use of the PA self-report questionnaire. An objective and more accurate method, such as an accelerometer, would be more reliable (*Meh et al., 2021*). Another limitation is that only the amount of sleep is measured, based on the subjects' self-report. The question of the amount of sleep was asked in a general way, without considering the context related to sleep quality. We did not take into account possible sleep problems in people with overweight and obesity, such as insomnia or snoring, which can significantly affect sleep quality and overall quality of life. For a more comprehensive assessment of the impact of sleep as a lifestyle factor, it would be worthwhile to include these aspects in future studies.

### Strengths and future research directions

FA is currently not recognised as a disorder. In this context, it is important that the study analysed a subgroup of people with overweight and obesity. It is also important that older people were studied. Finally, our results are based on actual metabolic and anthropometric measurements.

As the standard approach to weight loss of maintaining a healthy diet and exercise is often associated with ineffective adherence and overall weight gain, it seems reasonable to attempt to analyze the differences in physical and mental health in people with overweight and obesity (*Adams et al., 2019*). Their analysis provides a better understanding of the needs and challenges of treating people with overweight and obesity.

There is a case for multidisciplinary interventions in the care of people with overweight or obesity who want to lose weight, with a particular focus on their mental health. More longitudinal studies are also needed to assess the causal links between mental health and body weight.

For adults with overweight or obesity, careful selection of evidence-based treatment strategies is important. It may be prudent to investigate the effects of FA on weight loss and further incorporate addiction-specific treatment components to enhance weight loss and prevent weight regain in this subgroup (*Pietrabissa, 2018*; *Lawlor et al., 2020*). Although our study does not describe these interventions, we mention them as further avenues of research that may help people with overweight and obesity in general, and perhaps especially those with depressive symptoms and FA. If further research confirms that FA is a factor in the etiology of obesity, its treatment would be a supportive intervention for weight loss.

## CONCLUSIONS

A total of 22.7% of participants with overweight and obesity had symptoms of depression according to the SDS, and 18.6% met the criteria for FA according to YFAS 2.0. The older a person was, the less likely they were to have FA symptoms. We found that BMI, body fat mass, DBP and SB were statistically significantly higher in people with FA symptoms. Contrary to our assumptions, the presence of FA did not significantly differentiate the groups with respect to waist circumference, WHR, amount of visceral tissue, TC, HDL,

LDL, SBP, fasting blood glucose, comorbidities, total PA, eating behaviours, and frequency of consumption of different food groups.

Our findings complement the current literature on FA, particularly in older adults and metabolic parameters, and suggest further research directions. Future research should consider the development of new dietary assessment methods to assess the food intake of UPF in relation to FA.

Interdisciplinary interventions in the care of people with overweight and obesity who are trying to lose weight, with a particular focus on assessing their mental health, appear to be important. In line with other researchers and recommendations for the general population, we emphasise the need to reduce SB and highlight its association with weight management and mental health. Although our cross-sectional study design does not allow causal interpretations, increasing PA appears to be particularly important in the management of people with overweight or obesity and FA. This may be even more important than for people with depression alone, but future research is needed to explore these relationships further.

### Funding
The authors received no funding for this work.

### Competing Interests
The authors declare that they have no competing interests.

### Author Contributions
- Magdalena Zielińska conceived and designed the experiments, performed the experiments, analyzed the data, prepared figures and/or tables, authored or reviewed drafts of the article, and approved the final draft.
- Edyta Łuszczki conceived and designed the experiments, performed the experiments, analyzed the data, prepared figures and/or tables, authored or reviewed drafts of the article, and approved the final draft.
- Anna Szymańska performed the experiments, authored or reviewed drafts of the article, and approved the final draft.
- Katarzyna Dereń conceived and designed the experiments, performed the experiments, analyzed the data, prepared figures and/or tables, authored or reviewed drafts of the article, and approved the final draft.

### Human Ethics
The following information was supplied relating to ethical approvals (*i.e.*, approving body and any reference numbers):

The study was approved by the Bioethics Committee of the University of Rzeszów (Resolution No. 2023/07/0046 of 25 June 2023).

## Data Availability

The raw measurements are available in the Supplemental File.

## Supplemental Information

Supplemental information for this article can be found online at http://dx.doi.org/10.7717/peerj.17639#supplemental-information.

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
