# Peer review of "Food addiction and the physical and mental health status of adults with overweight and obesity"

_PeerJ, doi:10.7717/peerj.17639_

## Round 0.1 · original submission · Major Revisions

· Academic Editor

Major Revisions

Than manuscript is in need of revisions, as suggested by the reviewers

Reviewer 1 ·

Basic reporting

General writing could use some work, not only basic grammar but also the flow of the article as a whole.

Experimental design

Sufficient, though there seems to be some disconnect between objectives, hypotheses, analyses, and reporting of findings.

Validity of the findings

The novelty is not clear based on the manuscript

Additional comments

Here is line-by-line feedback:
Overall:
This is an interesting study but I think there is a lot of room to clean up your objectives and hypotheses to make sure they match what you actually analyzed in your stats. Making your intro, methods, results, and discussion all follow the same logical flow will make it a much easier and compelling read.
Introduction:
Lines 96:102: Make sure your objectives don’t reflect what you found but what you designed the study to assess. For example, you didn’t actually want to target adults starting at age 23 (based only our inclusion criteria), but that’s who you captured…
Lines 101-102: This phrasing is very confusing to me: “Assess the physical and mental health of the group on the basis of actual anthropometric and metabolic measurements and relevant questionnaires.” – based on your intro, why isn’t the independent variable here the presence of FA?
Lines 104-114: I think the wording of your hypotheses could use some modification based on the selection of your stats. For example, H1 suggests to me more of a correlation, per chi-square I would expect something more like: “determine differences in the prevalence of food addiction according to age”
Line 111: Right now H4 results are in the supplementary files – I would think that eating behavior is key element throughout your paper, you would want to include in main findings?
Methods
Throughout: Did you do any post-hoc analyses in you chi-squares? I would also think with this number of analyses you would need to control for multiple comparisons
Lines 122-129: I would think these sample data should go in results
Lines 127-129: This whole section could use some cleaning. Were these variables that describe your final sample or are these inclusion/exclusion criteria? Spelling out inc/exc criteria and moving descriptors of your final sample to results would help clarify
Lines 147-150: Did you use validated items for these variables when applicable (e.g., history of eating disorders)? If not, how did you go about adapting/creating items (even if they weren’t validated as part of the study – how did you develop them?)
Lines 155: It looks like there is a citation after the word “groups” but this does not match your citation style
Results
Throughout: I think your formatting of introductory sentences in paragraphs would be easier to read if you just opened with the result and put the table reference in parentheses after the sentence. For example, instead of: “We verified the relationship using a Chi-square independent test, the results of which are presented in the table below (Table 4),” you can say “We found a statistically significant difference in FA scores across age categories (Table 4)” (x2(2,N=172)…”
Also, I think you can more clearly tie your hypothesis to your results.
Line 328: This may be stylistic preference but your use of present tense reads awkwardly to me
Line 331: What normal value?
Line 332: What does this mean: “Taking into account the actual results of the physical health parameters,”?
Line 337: When you say “18.6% had FA symptoms according to YFAS 2.0” – does this mean they had ANY FA symptom or met criteria for FA?
Line 341: You describe the YFAS methodology but it’s not entirely clear to me how you operationalized your binary variable (FAS vs NFA) for analysis (e.g., is mild or severe dependence = FAS?)
Line 384-386: I think this detail should be in methods and indicated in your tables, but not here.
Lines 415-418: “Those who were overweight were by far the most likely to have neither FA nor depressive symptoms (70.6%), as were those with class I obesity (68.8%) and class II or III obesity (55.0%).” – those with obesity were most likely to have what? Also your use of the term “likely” is suggesting to me that you did regression, calculated probabilities, etc., but you are just describing them, correct? That may be my confusion. If you’re just describing the percentages, just say “The majority of individuals, regardless of weight, did not have FA or depressive symptoms”
Tables: Is there a way to condense your tables so they are easier to read? I’m not sure quite as much descriptive data is needed based on your hypothesis. Maybe just include mean and SD + analysis outcomes in one table per outcome?
Table 1: What is meant by “Increase” in food addiction? This is a cross sectional study so you are not measuring change.
Table 1: The cut-offs for some of your descriptors seem odd – there are a lot of nuances other than “retirement,” “occasional,” and “permanent employment”?
Discussion
Throughout: Laying our your discussion to match the order of your hypotheses and results would make it a much easier read.
Lines 439-444: This could use some massaging. Not wrong, but why include food addiction?
Lines 475-488: I think careful attention-to-detail is needed here when describing eating behavior versus dietary patterns and/or dietary intake. You seem to be using the term eating behavior to describe consumption of ultra-processed foods. As defined by the TFEQ (which you used), these are not the same. You don’t actually mention any TFEQ-related variables/constructs in this paragraph, and I would venture to say FFQs do not measure eating behavior.
Line 394: I would not say that 39% is “dominant”

Reviewer 2 ·

Basic reporting

- English proofreading recommended
- Please make use of non-discriminating language, e.g. people with obesity instead of obese people
- The rational of the manuscript should be more comprehensive. Why is it important to analyse FA in obesity? Given the high prevalence of obesity and overweight worldwide, it is necessary to develop effective treatment strategies that are tailored to individual needs. There might be a subgroup of people with overweight and obesity which fulfils the criteria for FA. If so, it is crucial to understand this subgroup in order to develop targeted interventions. You mention different aspects like eating styles, depression, physical activity and comorbidities which are associated with FA, but why is it important to analyse metabolic parameters? And based on that background: what’s the uniqueness of your study? The derivation of your hypotheses is not comprehensive and you should state the direction of relationship you expect based on the literature background (e.g. positive relationship between FA and depression).
- Please check for citation errors throughout the manuscript, e.g. line 67-69: According to Burmeister et al. (2013)…..
- Abstract: the reason for your study is not comprehensive and the conclusion should include deductions from your manuscript (results regarding FA).
- Abbreviations should be checked throughout the manuscript (e.g. FA and FAD, sedentary behaviour = SB, WHR should be spelled out the first time….)
- Tables: too many (see section below according to the shapiro tests and the overall number of parameters and analyses). The headings are wrong (table 5 -7)
- Methods: Please standardize the description of the instruments (e.g. the ratings 1-6 in brackets and the description italic, ranging from (1) never to…) and add psychometric properties, were applicable.
- How did you define FA and NFA? In the methods you describe different severities of FA, yet in the results you only describe FA vs. NFA

Experimental design

- I am not convinced that you used the proper analyses for your research questions. When describing relationships the use of correlation or regression analyses might be more comprehensive. That might also facilitate the interpretation of your results.
- I would recommend to remove the shapiro analyses from the manuscript and simply state in the methods, that parameters that were not normally distributed as measured by the SW test, were analysed by using non parametric tests.
- You should state if you used a correction method for multiple testing, like the Bonferroni correction.
- Why do you analyse FA and depression as one group?
- In addition, the large amount of analyses aggravates the understanding of your manuscript. I would recommend to maybe focus your analyses on less parameters, or groups of parameters.
- Please check for redundant information in the text body and the tables.

Validity of the findings

- Since FA is presently not recognized as a distinct disorder, research analysing this subgroup of people with overweight and obesity is of importance
- The cross-sectional design limits the interpretation of the results, yet it is of importance that especially older adults were analysed
- The huge amount of parameters and the selection of statistical methods aggravates the understanding of the results, but I am convinced that when focusing on fewer parameters based on a comprehensive background and a rethink of statistical analyses, the manuscript can fill a research gap, i.e. due to the metabolic parameters.

---

## Round 0.2 · Minor Revisions

· Academic Editor

Minor Revisions

After these minor revisions MS can be accepted

Reviewer 2 ·

Basic reporting

Thank you for the opportunity to re-review your revised manuscript. First, I’d like to thank you for your detailed response to my comments and your conscientious implementations of my suggestions. From my point of view, the manuscript substantially benefitted from the revision, i.e. due to the storyline of the article. I only have some suggestions for minor revisions:

Experimental design

see section below

Validity of the findings

see section below

Additional comments

General:
- Abbrevations: FA and FAD? Are there any differences? Please use continuously
- Still many tables. Please rethink necessity, i.e. table 3.

Abstract:
- l. 12: FA individuals should be changed into: individuals with FA.
- The background should include information (just one sentence) about why you implemented depression into your analyses
- Given your cross-sectional study design, you should be more careful in the interpretation of causal relationships (i.e. that SB causes FA/Depres). Just highlight the relationship and that when treating people with OB/OW with FA, the enhancement of PA seems to be particularly important (maybe even more than in people with depres. But future research is warranted).
- Abbreviations only when necessary in the abstract, e.g. FA. Do not include abbreviations in the abstract, that do not repeat in the abstract. Than introduce the abbreviations again in the main body.

Introduction:
- It might be easier to focus on the co-occurrence of FA and Dep, if you delete information about other comorbidities, like eating disorders. Otherwise, the question arises why you did not focus on FA and other EDs.
- Objectives: “The first was to determine the differences between FA and age, eating habits and physical health (comorbidities, metabolic and anthropometric parameters) of adults with overweight and obesity attending a private centre offering weight loss holidays for adults.” – differences between FA and age? I would recommend: Differences between FA and NFA concerning age, eating habits…
- H1: why do you expect age differences? What was the rational for grouping < 50 years, 51-65 and > 65?

Methods:
- L. 266: Please write out Food Addiction in the heading of the subdivision

Discussion:
- l. 487: “Both those with uncontrolled (66.1%), emotional (71.6%) and restrictive (58.7%) eating styles were more likely to be free of symptoms of FA and depression.” Please revise the sentence, since it is not comprehensive what “both” means.

---

## Round 0.3 · accepted · Accept

· Academic Editor

Accept

The manuscript can be accepted